# PestOn: An Ontology to Make Pesticides Information Easily Accessible and Interoperable

**Marco Medici** [1,*], **Damion Dooley** [2] and **Maurizio Canavari** [1]

1   Department of Agricultural and Food Sciences, Alma Mater Studiorum-University of Bologna, 40127 Bologna, Italy; maurizio.canavari@unibo.it
2   Centre for Infectious Disease Genomics and One Health, Simon Fraser University, Burnaby, BC V5A 1S6, Canada; damion_dooley@sfu.ca
*   Correspondence: m.medici@unibo.it

**Abstract:** Globally, present regulations treat pesticide use with a light touch, leaving users with scarce reporting requirements in the field. However, numerous initiatives have been undertaken to reduce risks from pesticide product use and provide the public with sufficient information. Nevertheless, food chain actors are not required to disclose much information on hazards, with many undervalued safety aspects. This situation has resulted in information gaps concerning the production, authorization, use, and impact of pesticide products for both consumers and regulatory stakeholders. Often, the public cannot directly access relevant information about pesticides with respect to retail products and their farm origins. National authorities have poor legal tools to efficiently carry out complete investigations and take action to mitigate pesticide externalities. We created the ontology PestOn to bridge these gaps and directly access pesticide product information, making existing data more useful and improving information flow in food value chains. This demonstration project shows how to integrate various existing ontologies to maximize interoperability with related information on the semantic web. As a semantic tool, it can help address food quality, food safety, and information disclosure challenges, opening up several opportunities for food value chain actors and the public. In its first version, the ontology PestOn accounts for more than 16,000 pesticide products that were authorized in Italy during the last 50 years and retrieved from the public pesticide register. The ontology includes information about active ingredients contained in pesticide products, roles, hazards, production companies, authorization status, and regulatory dates. These pieces of information can support agri-food stakeholders in classifying information in the domain of pesticide products and their active ingredients, while reducing unnecessary repetition in research. PestOn can support the addition of food attributes in the domains of human health, resource depletion, and eco-social impact, turning the spotlight on each possible improper use of pesticide products.

**Keywords:** pesticides; active ingredients; inputs; information; food production; ontology; RDF/OWL

## 1. Introduction

Pesticides are chemical agents used to protect crops from disease, pests, and weeds; improve plant growth; or inhibit the biological rhythm of flora and fauna that populate cultivated fields. Based on their role, pesticides are subdivided into various classes. Conventional pesticides comprise fungicides used to prevent fungi or their spores from affecting plants, insecticides used to kill insects, and herbicides used to limit weed diffusion in fields.

The massive use of pesticides (and fertilizers) in agriculture heavily increased with the advent of the third agricultural revolution at the end of the 1940s and resulted in rapid growth in agricultural production. However, their intensified use has resulted in serious implications for human and animal health and the environment. Numerous studies have proven that pesticide use damages flora and fauna both inside and outside cultivated fields, with cascading effects on humans [1–7]. Technically, pesticide application can be inefficient;

in China, spraying pesticides was estimated as being only 30% efficient, with most of the product being leached or driven away by wind and rain [8].

The rapid increase in pesticide use in the second half of the last century has led to increased attention on regulating pesticide products. Several policies such as rules, regulations, and taxes were introduced in particular in the 1990s to encourage appropriate pesticide use or limit pesticide application [9–14]. Numerous initiatives were undertaken to reduce risks from pesticides and provide the public with clear information on product labels. Where adopted, pesticide regulations are stringent: all pesticide products must be registered, and their use permitted for (or restricted to) professional users by national regulatory authorities. However, exceptions exist in most countries. For instance, pesticides that have been banned in Europe and North America are still in use in Africa, Asia, and South America [15].

Pesticide producers are not required to disclose much information on chemicals' hazard traits. In the U.S. and in the EU, nondisclosure includes data evaluation records, product inert ingredients, product ingredient source, product chemistry registration data, sales, production, or other commercial information ([16]). In the EU, the public can only indirectly access relevant information about pesticides by addressing the national authority, which in turn can request information from the producer ([17]). This situation has produced chemical data gaps, with many aspects being undervalued relative to the chemical role (function), price, and performance of pesticide products. This lack of information undermines safety, because national authorities cannot rely on the necessary legal tools (as well as on funding and resources) to efficiently carry out complete investigations and take action to mitigate externalities of hazardous agrochemicals [18].

Turning to pesticide use, pesticide regulations oblige professional users (in most cases, the farmers themselves) to maintain detailed records regarding the application of pesticides. Typical application records require the specification of the name of the pesticide product, the surface covered, the crop where the pesticide product was used, the dose of application, the date and location of the application, the name of the operator, and the purpose of application. In the EU, records must be kept for a minimum of three years after application, while in the U.S., this period is two years. Apart from the fact that pesticide damage can often occur over a much longer period, it is a common opinion that present regulations treat pesticide use with a light touch, leaving users with scarce reporting requirements [19–21]. For example, in the absence of a medical emergency, farmers and other landowners adjacent to fields where pesticides are applied have no means to determine whether chemicals are being applied safely, which would minimize the risk of human damage ([19,22]). A virtuous example of open data concerning pesticide use is represented by California's pesticide-use reporting system, which can serve as a model for future pesticide disclosure programs. In California, all the data from pesticide applications are reported in an open database managed by the California Department of Pesticide Regulation [23]. This approach has enabled the identification of health effects from residential exposures to pesticides, the monitoring of surface water pollution, the determination of honeybee pesticide exposure, and the safeguarding of biodiversity, locating mapping sites occupied by endangered species and avoiding nontarget exposures. The Californian case, which constitutes a unicum in the world, represents a significant example of how the safeguarding of the environment and human health involves, by necessity, full information disclosure. Nevertheless, beyond this example, lack of direct access to information about pesticide use is a recurrent issue elsewhere, along the lines of the aforementioned lack of access to pesticide production data. Similar levels of information disclosure and analyses are not possible in other countries [24].

Nevertheless, making information about pesticide characteristics available to the public and improving pesticide-use reporting systems would help prevent human and environmental harm and make progress toward a reduction in pesticide misuse by providing an incentive for professional users to perform responsible pesticide use. The appropriate use of pesticides requires accessible and easy-to-implement tools to perform sustainability assessments, including ecological risks [25]. For these reasons, our ultimate goals are to

provide better metrics to monitor pesticide information, allow access to comprehensive application information, and allow interoperation across the domains of food value chains, food safety, human health, and environmental impacts. We also aim to provide a tool to measure the appropriateness of the numerous regulatory efforts at the global level.

Specifically, in the domain of pesticides, the potential of semantic resources can help precisely define the numerous attributes of pesticide products emerging from production and authorization processes and, more importantly, from their use in a comprehensive information framework in the form of an ontology. An ontology is a representation of knowledge bases by means of a formalized domain model, specifically with linked open data (LOD). It can assure dynamic interoperability (dynamic interoperability can be defined as syntactic interoperability (i.e., using the same protocol or format) plus semantic interoperability (i.e., the ability to automatically interpret data) and support the rapid traceability of products and inputs. For these reasons, semantic resources represent an opportunity to address the sustainability challenges of value chains, especially in the food and agriculture sector [26,27].

Particularly in the domain of pesticides, semantic resources can be useful for compliance purposes as they meet several requirements that are needed to make pesticide-use data incorporated into environmental and health risk assessments, as reported by [24]. They can (i) allow datasets to be published and downloadable with a user-friendly online interface; (ii) help professional users with digital submission, with additional options to maximize compliance; and (iii) support reporting, including details of pesticide products applications (active ingredients and their concentration, rate and timing of application, and target crop variety).

Semantic resources can be used to gain new knowledge about the patterns and trends of actual pesticide use. A pesticide ontology can represent a strategy to open data in the domain of agricultural inputs and achieve intradomain interoperability. This can be the first step to link agricultural operations across food value chains more efficiently and effectively, with several advantages. For example, supply chain actors can benefit from easier standardized processes to comply with regulated farming approaches (e.g., organic farming and integrated farming); consumers can track back the amount of food production inputs used, thus verifying information of where, when, and how food was produced. Additionally, the maximization of compliance can ease third party evaluation mechanisms such as food safety regulations, life-cycle assessments, and ecological risk assessments, to name but a few.

Given the barriers associated with accessing pesticide information and the lack of standardized research and regulatory pesticide data, a new approach to the use of pesticide information is presented in this paper. This paper describes an ontology for the domain of pesticide products so that their characteristics and features can be easily accessed, interoperable, and jointly usable by food system stakeholders. The Italian Ministry of Health pesticide authorization framework is represented in the their public pesticide register, a database of over 16,000 pesticide products registered during the last 50 years, which contains several key production and authorization attributes. This database was transformed into an equivalent ontology schema, PestOn, along with import files containing an RDF graph database of all the same pesticide data.

## 2. Materials and Methods

Some subsequent steps were adopted in this study to make pesticide data accessible and semantically interoperable. First, the public pesticide register dataset was exported and saved in a structured, nonproprietary format (.tsv). Thus, an ontology scheme was designed to represent relations between concepts and entities based on relevant information about pesticide products available in the dataset field specification. Then, a review of existing ontological relations and shared terminology was conducted to cover the need to assemble an ontology of pesticide products; this process was also necessary to determine possible correspondences between concepts already defined in other ontologies. The OBO Foundry

framework approach was adopted [28,29] to create a language of relations and entity types that are common to the entire OBO Foundry encyclopedia, favoring the reuse of existing relationships and entities as much as possible. Finally, the ontology was implemented by means of a set of ontology editors able to encode the model in the web ontology language (OWL), which is the referenced format for ontologies.

### 2.1. Information Alignment

Pesticide products comprised 16,458 items registered up to 14 October 2021 by the Italian regulation authority (Ministry of Health; the registry is openly accessible at http://www.fitosanitari.salute.gov.it (accessed on 21 May 2022)). For each item, the following attributes were reported:

a. Commercial name of the pesticide product;
b. List of active ingredients contained therein (name and concentration);
c. Chemical role (e.g., fungicide, insecticide, plant growth regulator, adjuvant, herbicide, etc.);
d. Type of associated hazard(s) (e.g., dangerous for the environment, flammable, toxic, etc.);
e. Product formulation (physical feature);
f. Producer company name;
g. Location of the producer company (city);
h. Legal and administrative address of the producer company;
i. Parallel importing. Parallel importing refers to branded goods that have been manufactured by or are under license of the brand owner, but may have been formulated or packaged for a particular market and are imported into a different jurisdiction in contradiction to the brand owner's intention, thereby sold at a reduced price [30]. Parallel importing is regulated differently between countries (IP) and used by nonprofessional users: PFnPO and PFnPE (Italian legislation (Law decree 22/01/2018 n.33) distinguishes between the attribute PFnPO, concerning products that can be used on ornamental crops, such as those cultivated in homes, balconies, and home gardens; and PFnPE, which refers to products that can be applied to edible crops in vegetable gardens and small orchards ($<500$ m$^2$), or small vineyards, olive groves, and cereal cultivations ($<5000$ m$^2$). http://www.gazzettaufficiale.it/eli/id/2018/04/16/18G00058/sg (accessed on 21 May 2022));
j. Issue date of the pesticide product;
k. Expiry date of the pesticide product;
l. Revocation date of the pesticide product;
m. Revocation decision date of the pesticide product;
n. Revocation reason;
o. Authorization status of the pesticide product (e.g., authorized, revoked, reregistered).

In detail, the commercial pesticide name (a) was considered as the main pattern. Then, other distinctive and repetitive invariants across the initial pesticides dataset were identified: active ingredient (b), chemical role (c), hazard (d), and formulation (e). Each of these patterns were carefully checked with, in order of importance, existing ontologies or other types of linked open data, or other existing standards, as the following details.

The bulk of active ingredients (b) was automatically mapped with Zooma (http://www.ebi.ac.uk/spot/zooma/ (accessed on 21 May 2022)) to CHeBI ontology (http://www.ebi.ac.uk/chebi/ (accessed on 21 May 2022)), a nonproprietary freely available dictionary of molecular entities, including synthetic products such as drugs or chemicals used to intervene in the processes of living organisms; active ingredient concentrations were imported as numerical values with URLs for units of measure matched with units-of-measurement (UOM) ontology (http://units-of-measurement.org/ (accessed on 21 May 2022)). CHeBI was also used to match the chemical role (c) associated with pesticide products.

Symbols, codes, and mention of hazard (d) were found to be in accordance with the EU legislation ([31]), in turn aligned with the Globally Harmonized System of Classification

and Labelling of Chemicals (http://unece.org/ghs-implementation-0 (accessed on 21 May 2022)) (GHS). Product formulations and descriptions (e) were manually matched with labels for formulation codes provided by CropLife International (CropLife International is an international trade association of agrochemical companies founded in 2001) [32]. In addition, formats were matched against major product uses provided by IPARC (The International Pesticide Application Research Consortium (IPARC), Department of Biology, Imperial College, London), a higher level of summarization compared with CropLife International standard.

Producer company name (f) was imported under the NCIT schema (http://www.ontobee.org/ontology/NCIT?iri=http://purl.obolibrary.org/obo/NCIT_C54131 (accessed on 21 May 2022)), while producer company locations (g) were matched with location patterns in Wikidata (http://www.wikidata.org/ (accessed on 21 May 2022)) entities such as 'town' (wd:Q3957), 'city' (wd:Q515), and 'municipality' (wd:Q15284); the resulting Wikidata entities were obtained through Wikidata Query Service in SPARQL. The remaining locations were manually searched; finally, location labels were normalized and polished (e.g., initial letters were capitalized and accents were corrected).

Producer company addresses (h) were schematized with the PostalAddress structure available at Schema.org (http://schema.org/PostalAddress (accessed on 21 May 2022)). IP, PFnPO, and PFnPE, which are binary attributes in source data, were converted into classes of product (i). Issue, expiry, and revocation dates (j, k, l, and m) were interpreted as classes of dates having a legal significance; thereby, they were imported as instant entities (http://www.w3.org/TR/owl-time/#time:Instant (accessed on 21 May 2022)) with a date format. Revocation reason (n) and authorization status (o) were simply imported as strings.

CHeBI and Wikidata labels were added to the initial dataset using ad hoc designed R scripts [33]. R scripts were also used to preprocess the initial dataset and import information with ROBOT [34].

### 2.2. Ontology Implementation

The ontology about pesticides, named PestOn, was implemented using ROBOT template, an ontology editor able to encode the model in OWL. It consists of a template-driven process that is able to synchronize information between tabular data and existing semantic resources, in the form of OWL triples. In this process, the initial dataset was progressively imported into the ontology using OntoFox [35], a web-based ontology tool that fetches terms and axioms respecting ROBOT template to populate the ontology. As a final step, the ontology was uploaded into Protégé to verify the syntax correctness and possible undesired results from the reasoning.

### 3. Results

### 3.1. Alignment Outcomes

Concerning pesticide active ingredients, more than 99% of the entities were successfully matched against CHeBI ontology, with a few exceptions. Primarily, some active ingredient terms did not match because they refer to organic entities distinct from chemical compounds, such as fungi and bacteria (e.g., *Bacillus*, *Trichoderma*, *Spodoptera*, etc.). Secondly, several active ingredients (bromopropylate, triazbutil, sulfosulfuron, pinolene, copper oxalate, and chlorflurenol), including their synonyms, were not tracked in CHeBI. This issue was reported to the editor EMBL-EBI. Wikidata mapping returned all matches, which meant that every location in the initial dataset was mapped as a Wikidata term label. Concerning other alignments, no inconsistency was found.

### 3.2. PestOn Ontology

PestOn ontology is openly available at the following URL: http://github.com/marco-medici/peston (accessed on 21 May 2022), in a GitHub repository to support cross-domain interoperability, traceability, and reuse. The PestOn scheme is shown in Figure 1. The ontology contains 1957 classes, 15 object properties, 6 data properties, and a total of 100,350 individual instances. In its first version, PestOn involved 795 active ingredients contained in 16,458 pesticide products, with IDs based on the assigned IDs of the original database. Active ingredients were linked to pesticide products through the relation 'active ingredient in'. In total, 532 producer companies were modeled. In PestOn, all items (a–o) and their subclasses were provided with a literal definition.

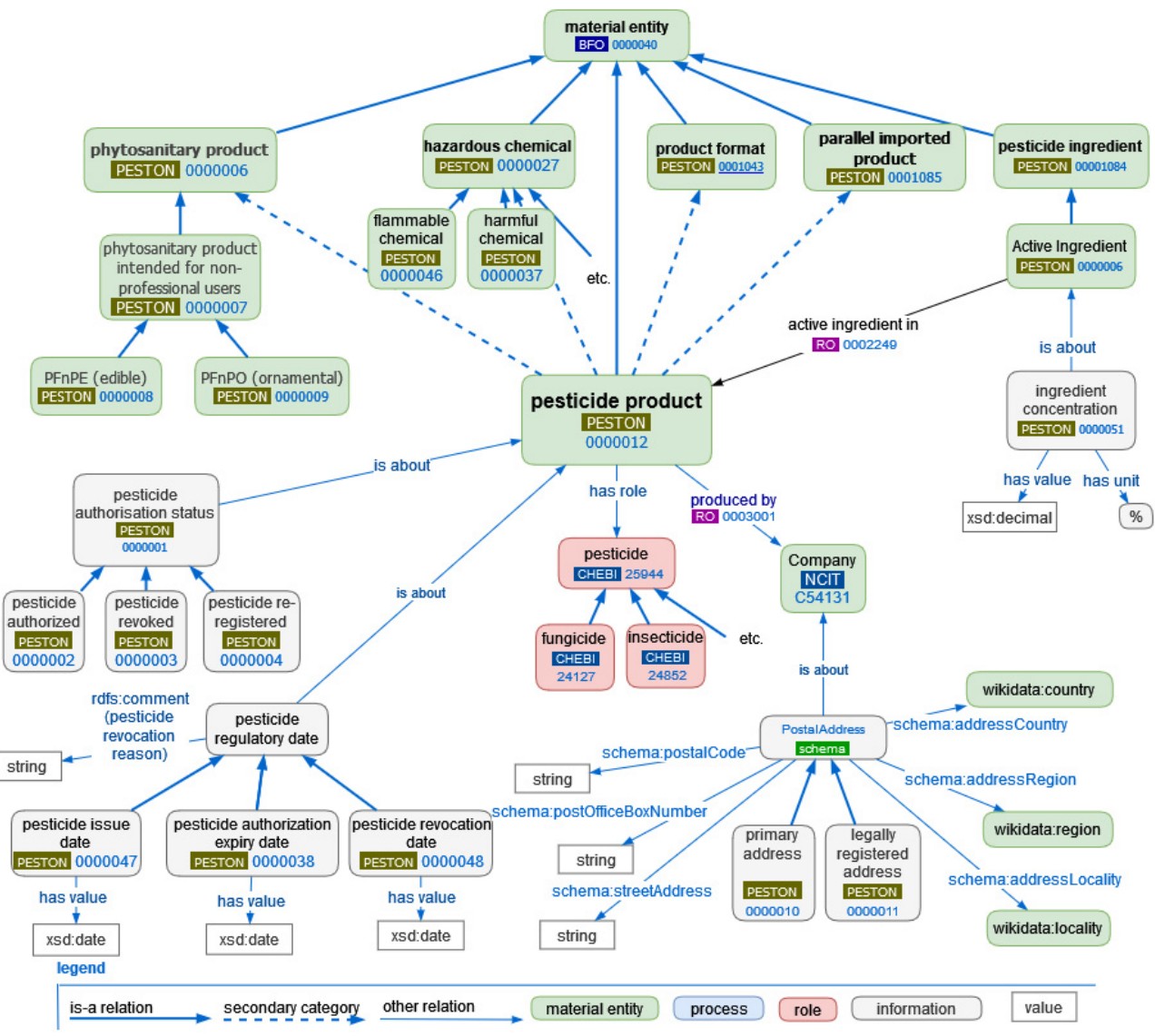

**Figure 1.** The PestOn scheme of relations with entities, processes, and information.

A sample pesticide product containing two active ingredients is displayed in Figure 2, with available features and characteristics (the company's legally registered address was omitted for clarity). In PestOn, the information describing a certain entity can be aggregated across multiple dimensions. For instance, an overview of the current available products for each role can be obtained, as well as a summary of products containing a given active ingredient. Figure 3 shows which pesticide products contain a selected active ingredient, with information limited to product authorization. Dates were modeled as data properties of pesticide regulatory events (issue, authorization, and revocation). Although the pesticide authorization status field can be intuitively inferable from the pesticide regulatory dates, we decided to explicitly state the authorization status because other date event types were missing, particularly those regarding a number of pesticide products issued before 2010.

PestOn ontology captures, stores, and provides information needed by food system stakeholders for numerous purposes. PestOn can support professional users with digital submission of farm registers and reporting details of pesticide product applications, maximizing compliance. Pesticide data can also be aggregated to other ontologies contributing to defining a more comprehensive view of how agricultural operations are linked across food value chains and how food inputs may affect environmental and health risk assessments.

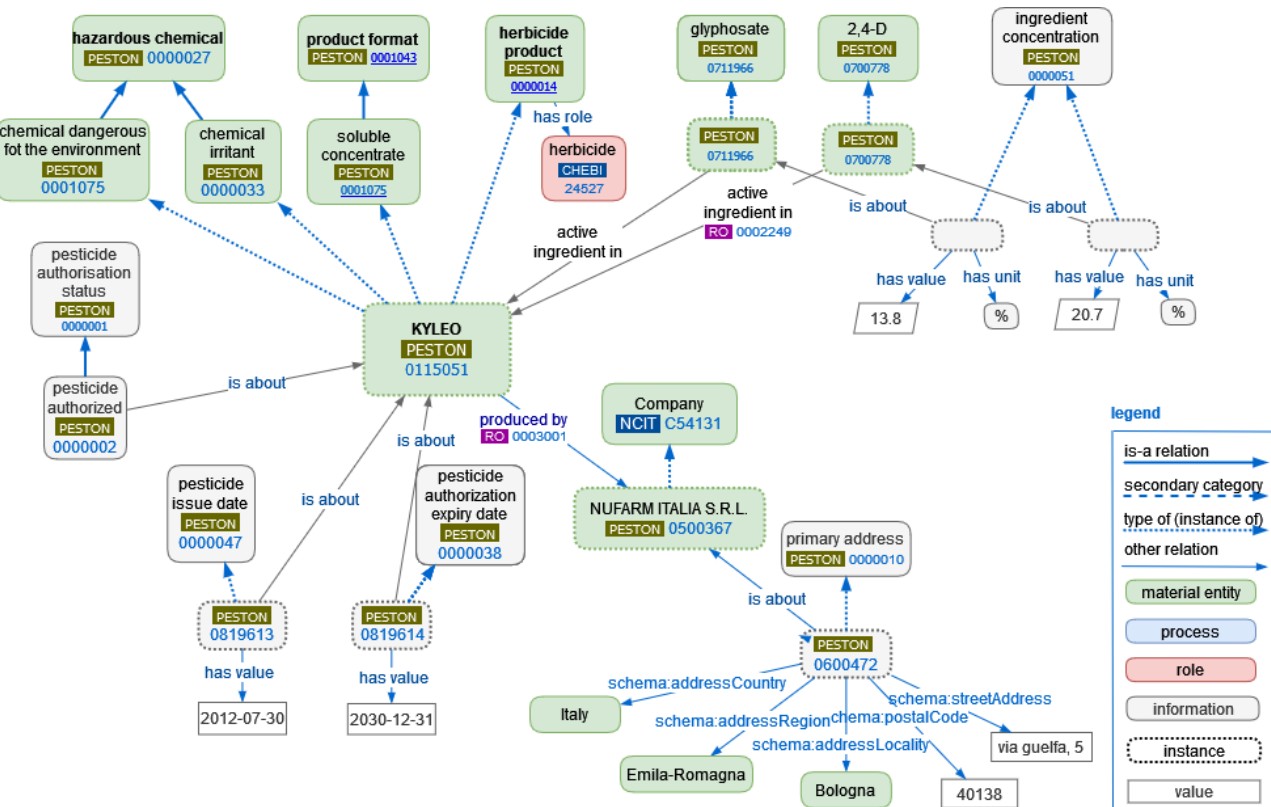

**Figure 2.** Overview of a sample herbicide product (KYLEO).

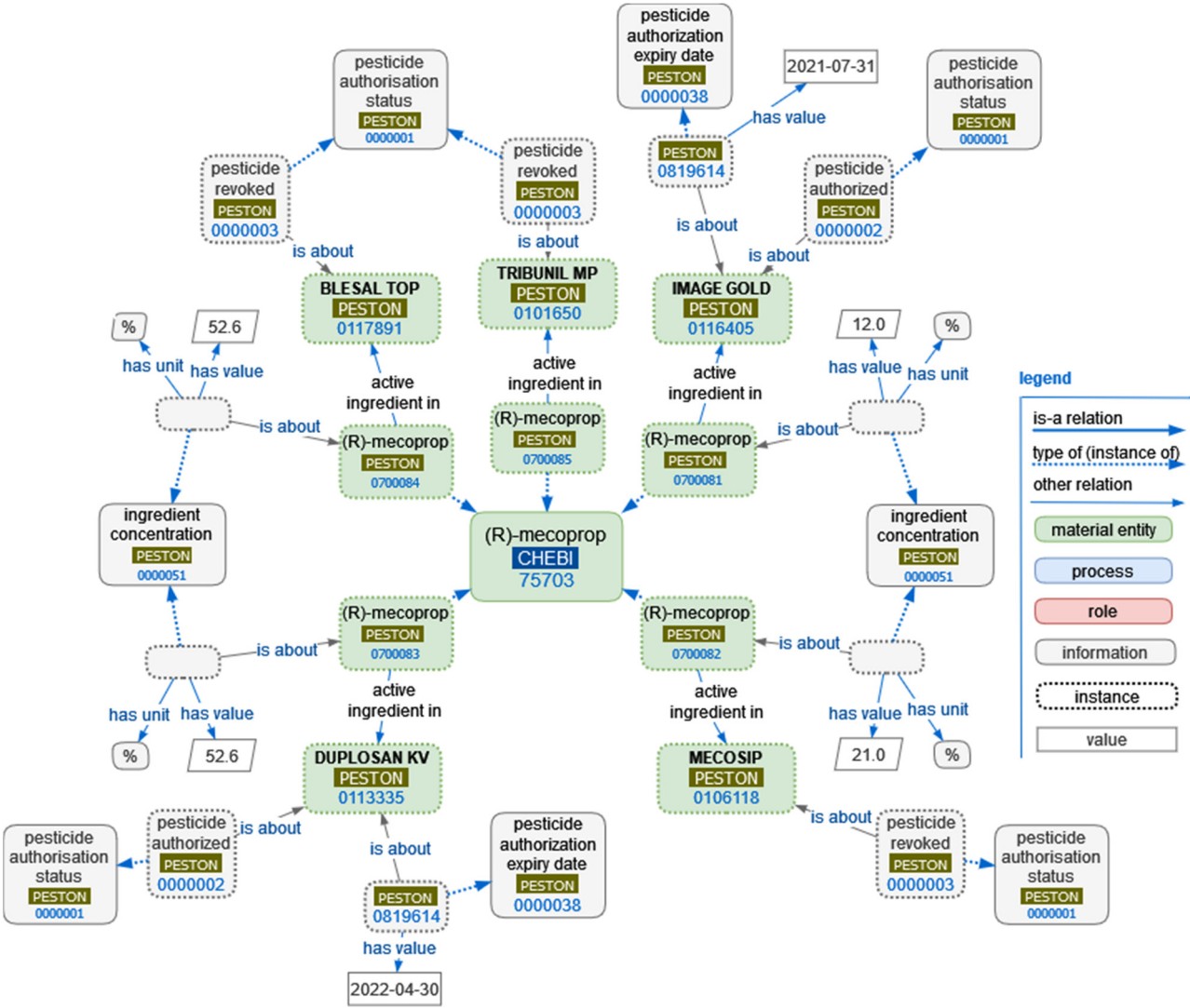

**Figure 3.** Pesticide products containing the active ingredient (R)-mecoprop, with information about product authorization status.

## 4. Discussion

PestOn is an open-access ontology of pesticide products aiming to advance knowledge in the domain of agricultural inputs and provide greater transparency on pesticide characteristics. It is characterized by high-quality pesticide data, which may benefit food value chains. The knowledge base represented in PestOn reveals pieces of information that can support agri-food stakeholders in classifying information they gather and types of information they seek in the domain of pesticide products and their active ingredients, while reducing unnecessary repetition in research. PestOn prototypes the domain of pesticides based on the Italian authorization framework, with the opportunity to be applied to other countries. The term 'pesticide authorization status' is defined generally so that it can pertain to other jurisdictions. New, more specific semantic variations can be introduced to cover the authorization processes of other jurisdictions. The subordinate classes of 'pesticide authorization status', i.e., pesticide authorized, revoked, and reregistered, might be pertinent to only some countries; in this regard, to make generalizations about authorization status, it would be helpful to know more about the way authorization processes and dates are standardized in other jurisdictions.

PestOn can support a range of SPARQL queries to verify the use of pesticide products, from simple eligibility checks that prevent the use of expired products to performing



more sophisticated inquiries aimed to assess compliance with the various types of farming approaches (e.g., organic standard, integrated pest management, etc.). PestOn can support the addition of food attributes in the domains of human health, resource depletion, and eco-social impact, turning the spotlight on each possible improper use of pesticide products.

To date, efforts aimed at providing a comprehensive information framework in the domain of pesticide products have been missing. Many initiatives in the field of semantic technologies applied to food value chains have led to partial results. This is due to multiple reasons: for one, proprietary schemes are often based on nondisclosure of data; in addition, proprietary metadata and resource identifiers undermine full data sharing and interoperability. In addition to these technical reasons, other issues lie in the fact that comprehensive frameworks covering enough food aspects (such as nutritional, economic, environmental, and social) require high expertise across multidisciplinary backgrounds. Private and public sector efforts often fail to coordinate data harmonization efforts. There are also considerations related to privacy, ethical use of data, market control, fraud, and policy that require stakeholder contribution [26].

For these reasons, to date, ontologies or other frameworks based on linked open data are scarcely documented in the literature, with only a few exceptions represented by studies seeking dynamic interoperability in food systems. Two relevant examples are FoodOn and the Food System Dashboard. FoodOn is a consortium-driven project aimed at building a comprehensive farm-to-fork set of ontologies describing foods from various cultures worldwide, addressing terminology gaps and supporting traceability [36]. The Food System Dashboard describes food systems by bringing together data across over 140 indicators from over 30 sources, showing how food systems data can be visualized and compared across multiple global scales and domains [37].

Compared with these efforts, PestOn occupies a complementary space, trying to enhance the quality and quantity of shared information concerning primary food production (e.g., the cultivation of cereals, fruit, and vegetables), which is accompanied by large pieces of information generated during agricultural operations. These data, usually stored in farm registers, are thus shared with other food supply chain actors or public administrations for ritual compliance checks. Semantic resources encompassed in PestOn can help food supply chain actors comply more easily with regulations, and prepare the ground for further development of tools and applications for data traceability in the food value chains. In an optimistic view, consumers may also benefit from such information describing where, when, and how food is produced, simply by tracking back types and amounts of input used in production processes. PestOn can efficiently support the tracking of pesticides use, facilitating the European Green Deal goal of reducing pesticide use by 50% by 2030 and promoting a relevant change toward a more sustainable agri-food system.

One of the key strengths of semantic technologies is that they are extremely versatile. They can be applied to various contexts. For instance, PestOn can be easily extended, enriched with additional product data from other national pesticide registers, and translated into every language. PestOn aims to enhance dynamic collaboration and user-driven experimentation within food systems; information retrieved from existing pesticide registers may be more easily accessible, interoperable, and jointly usable by food system stakeholders. Equitable access to better quality data and assessment capabilities for use by food value chain actors can, in turn, facilitate the cocreation of innovative solutions and potential new avenues for future research. In addition, such tools can also motivate the exploration of cleaner chemical technologies, lowering market barriers to green chemistry.

Future research can explore connections with other domains, such as human and environmental exposure to stressors, for instance, taking advantage of the dedicated ontology ECTO (http://www.ebi.ac.uk/ols/ontologies/ecto (accessed on 21 May 2022)). Literature about pesticides is generally divided into two main lines, i.e., health literature focusing on individuals directly exposed to pesticides (e.g., farmers), and literature on preference valuation that has mostly focused on those with indirect exposure such as consumers [38]. PestOn may be beneficial to both areas. On the one hand, it can support investigations

about productivity–health tradeoffs that are motivated in occupational settings [39,40], and it can be easily matched with tools for the estimation of the external costs of pesticide use (e.g., [41,42]. On the other hand, by representing pesticide usage facts, it can help consumers make wise choices related to the food that they purchase [43].

## 5. Conclusions

Modern data science tools such as semantic technologies have the potential to overcome information gaps that today often characterize agriculture and food systems. Information concerning the production, authorization, use, and impact of pesticide products is currently subjected to scarce reporting requirements, nondisclosure, or poor emphasis on safety aspects. Aimed at bridging these information gaps, PestOn was created to better describe the domain of pesticide products so that their characteristics and features can be easily accessible, interoperable, and jointly usable by food system stakeholders, making existing data more useful. PestOn involves 795 active ingredients contained in 16,458 pesticide products, linked by a formalized domain of linked open data. As a starting point to build the ontology, the public pesticide register issued by the Italian Ministry of Health, including products described by several attributes from the production and authorization domains, was leveraged. PestOn can easily be enriched with additional terms and data from other national pesticide registers and jurisdictions, with the aim of modeling what is in common in global pesticide management processes. Future research may explore connections with the domains of agricultural production, environmental impacts, and nutritional aspects. Further technological developments may represent facts around pesticide usage, helping consumers to make wise choices related to the food that they purchase.

**Author Contributions:** Conceptualization, methodology, software, formal analysis, and investigation: M.M. and D.D.; data curation and writing—original draft preparation: M.M.; resources and writing—review and editing: M.M., D.D. and M.C.; validation and visualization: D.D. All authors have read and agreed to the published version of the manuscript.

**Funding:** This research received no external funding.

**Institutional Review Board Statement:** Not applicable.

**Informed Consent Statement:** Not applicable.

**Data Availability Statement:** The work described in this paper uses data and information in the public domain, with no privacy data involved. Therefore, the level of information disclosure and analyses are not a source of concern. PestOn ontology is openly available at the following URL: http://github.com/marco-medici/peston (accessed on 21 May 2022).

**Acknowledgments:** Marco Medici sincerely thanks Aldo Gangemi for introducing him to knowledge engineering modeling, and Valentina Presutti and Andrea Giovanni Nuzzolese for their kind availability and patience during their lectures.

**Conflicts of Interest:** The authors declare no conflict of interest.

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
