# Peer review of "PestOn: An Ontology to Make Pesticides Information Easily Accessible and Interoperable"

_sustainability, doi:10.3390/su14116673_

Round 1

Reviewer 1 Report

This article is on an important topic with significant implications for several stakeholders. The most crucial element in a possible new version of the article would be to better position your article and especially your contributions to the literature. Here are my comments on the essential points to improve.

The abstract of the article is incomplete. Information about the data, the results, and the contributions is needed. The reader can’t understand the relevance of this article to pesticide research.

The different sections of the article also need to be reviewed to ensure that the objective is presented coherently. I do not see your contribution regarding "better metrics to monitor pesticide information.” To me, the current version of the article shows a promising initiative. Still, its contribution to pesticide management needs to be reviewed to justify the publication of this case. I do not underestimate the work required to develop this platform and the social benefits of this project. I question the scientific value of the findings. The discussion of the results needs to engage more with the literature on pesticide management. In addition, there should be a clear section on contributions, limitations, and avenues for future research. For example, future avenues of research should be explored in greater depth and not remain at a superficial level.

Finally, it is tough to understand the steps taken to complete this project. The first section of the methodology must be reviewed in depth to clarify your approach concerning your research objective.

I hope you find these comments helpful.

Author Response

Thank you for your feedback. We have adjusted the manuscicript considering all of your suggestions.

Reviewer 2 Report

This paper is much more professional than a scientific article. The idea presented in this article is not particularly original either because such tools already exist. Still, the paper is written correctly and will surely find readers. If the Editorial Board has already accepted the topic of this article, as a reviewer I accept its publication in this form.

It is interestingly written, relevant, well written and relevant to the content. My doubts are whether this is a professional or an original scientific article.
Given the above qualities, I suggest you publish it in its current form. 

Author Response

Thank you very much for your feedback!

Reviewer 3 Report

The paper is well organized and easy to read. The design method used to build the ontology is given in a concise way in §2. The selected examples from Figures 2 and 3 illustrate well the type of information contained in the ontology and its possible use. The Discussion Section gives the reasons why this ontology is a contribution to improve data integration and sharing information about pesticide characteristics, aiming to support the tracking of pesticide use and facilitate the transition to a more sustainable agri-food system. 

Some minor syntactic errors:

  • lines 155 and 166: align with the rest of the list's items
  • line 155: "and" is strange here

Other remarks:

  • line 194: what is the meaning of "polished" in this context?

Author Response

Thank you very much for your feedback!

Round 2

Reviewer 1 Report

The problems mentioned in the previous version are still present. Contributions to the literature are still unclear. Changes to the article are superficial and not significant, as requested. The paper needs to be better grounded in the literature and clarify its contributions. 

Author Response

Please find attached our reply.

Round 3

Reviewer 1 Report

It was relatively simple to integrate your case into some of the literature on pesticide management—for example, pesticide information reporting and stakeholder knowledge. I intended to leave you with the direction you wish to take the paper. Given the nature of your response, I will have no choice but to recommend rejection. For me, the mere description of a case is not a sufficient scientific contribution. After that, I will leave it to the editor to decide. 

Here are some articles that could be incorporated into your paper on pesticide communication and knowledge:  

Dara, S. K. (2019). The new integrated pest management paradigm for the modern age. Journal of Integrated Pest Management10(1), 12.

Escalada, M. M., & Heong, K. L. (2007, September). Communication and implementation of change in crop protection. In Ciba Foundation Symposium 177Crop Protection and Sustainable Agriculture: Crop Protection and Sustainable Agriculture: Ciba Foundation Symposium 177 (pp. 191-207). Chichester, UK: John Wiley & Sons, Ltd..

Allen, W., Ogilvie, S., Blackie, H., Smith, D., Sam, S., Doherty, J., ... & Eason, C. (2014). Bridging disciplines, knowledge systems and cultures in pest management. Environmental Management53(2), 429-440.

Escalada, M. M., & Heong, K. L. (2004, September). The case of using mass media: Communication and behavior change in resource management. In New directions for a diverse planet. Proceedings of the 4th International Crop Science Congress (Vol. 26).

Escalada, M. M., & Heong, K. L. (2004, September). The case of using mass media: Communication and behavior change in resource management. In New directions for a diverse planet. Proceedings of the 4th International Crop Science Congress (Vol. 26).

Author Response

Dear Reviewer, thank for your suggestions. Please find attached our comments.
